# The effects of varying drought-heat signatures on terrestrial carbon dynamics and vegetation composition

Elisabeth Tschumi[1,2], Sebastian Lienert[1,2], Karin van der Wiel[3], Fortunat Joos[1,2], and Jakob Zscheischler[1,2,4]

[1]Climate and Environmental Physics, University of Bern, Bern Switzerland
[2]Oeschger Centre for Climate Change Research, University of Bern, Bern, Switzerland
[3]Royal Netherlands Meteorological Institute, De Bilt, The Netherlands
[4]Department of Computational Hydrosystems, Helmholtz Centre for Environmental Research – UFZ, Leipzig, Germany

**Correspondence:** Elisabeth Tschumi (elisabeth.tschumi@unibe.ch)

**Abstract.** The frequency and severity of droughts and heat waves are projected to increase under global warming. However, the differential impacts of climate extremes on the terrestrial biosphere and anthropogenic $CO_2$ sink remain poorly understood. In this study, we analyse the effects of six hypothetical climate scenarios with differing drought-heat signatures, sampled from a long stationary climate model simulation, on vegetation distribution and land carbon dynamics, as modelled by a dynamic global vegetation model (LPX-Bern v1.4). The six forcing scenarios consist of a *Control* scenario representing a natural climate, a *Noextremes* scenario featuring few droughts and heatwaves, a *Nocompound* scenario which allows univariate hot or dry extremes but no co-occurring extremes, a *Hot* scenario with frequent heatwaves, a *Dry* scenario with frequent droughts, and a *Hotdry* scenario featuring frequent concurrent hot and dry extremes. We find that a climate with no extreme events increases tree coverage by up to 10 % compared to the *Control* and also increases ecosystem productivity as well as the terrestrial carbon pools. A climate with many heatwaves leads to an overall increase in tree coverage primarily in higher latitudes, while the ecosystem productivity remains similar to the *Control*. In the *Dry* and even more so in the *Hotdry* scenario, tree cover and ecosystem productivity are reduced by up to -4 % compared to the *Control*. Regionally, this value can be much larger, for example up to -80 % in mid-western U.S. or up to -50 % in mid-Eurasia for *Hotdry* tree ecosystem productivity. Depending on the vegetation type, the effects from the *Hotdry* scenario are stronger than the effects from the *Hot* and *Dry* scenario combined, illustrating the importance of correctly simulating compound extremes for future impact assessment. Overall, our study illustrates how factorial model experiments can be employed to disentangle the effects from single and compound extremes.

## 1 Introduction

The terrestrial biosphere sequesters about 30 % of the anthropogenic $CO_2$ emissions (Friedlingstein et al., 2020). Different factors such as increasing atmospheric $CO_2$ concentrations, higher temperatures, or, on a more regional scale, water or nutrient availability, can increase or decrease the terrestrial carbon sink. Different biomes may also react differently. While warmer temperatures are likely to increase productivity in high latitudes and altitudes due to an increase in the growing season length,

productivity may be reduced in warmer regions because of higher evaporation and stomatal closure (Friend et al., 2014). Overall, there is evidence that the vulnerability of trees to hotter droughts may increase but this may also be compensated by higher $CO_2$ concentrations and associated increased water use efficiency (De Kauwe et al., 2013). However, future projections of the terrestrial carbon sink remain highly uncertain (Friedlingstein et al., 2014).

A potentially large contribution to the uncertainty in carbon cycle response to climate change may stem from the impacts of climate extremes. Climate extremes can cause devastating impacts on the natural environment (IPCC, 2012; Reichstein et al., 2013; Frank et al., 2015; von Buttlar et al., 2018; Senf et al., 2020). At the same time, extreme impacts are often not linked to single climate extremes but to a combination of anomalous drivers (Zscheischler et al., 2016; Flach et al., 2017; Pan et al., 2020; Tschumi and Zscheischler, 2020; Van der Wiel et al., 2020; Vogel et al., 2021), also called compound events (Zscheischler et al., 2018, 2020).

Arguably, drought and heat are among the most damaging hazards to terrestrial vegetation (Allen et al., 2010; Reichstein et al., 2013; Zscheischler et al., 2014b; Frank et al., 2015; Sippel et al., 2018; von Buttlar et al., 2018; Senf et al., 2020). In many cases, drought and heat predispose or interact with other hazards and disturbances such as forest fires and insect outbreaks (Seidl et al., 2017). In particular, an increasing occurrence of warm droughts has already lead to increased vegetation impacts on northern hemispheric ecosystems over the observational period (1982-2016, Gampe et al., 2021). However, differentiating impacts between drought and heat alone and compound drought and heat remains a challenging task. Disentangling these impacts is important, as co-occurring droughts and heatwaves tend to have larger impacts compared to the sum of impacts from droughts and heatwaves separately (Zscheischler et al., 2014b; Ribeiro et al., 2020), for example because a drought exacerbates the impacts of a heatwave through reduced evaporative cooling (Yuan et al., 2016). Furthermore, projections of droughts and heatwaves can differ strongly across different climate models (Herrera-Estrada and Sheffield, 2017; Zscheischler and Seneviratne, 2017).

The impacts of climate extremes on vegetation and the terrestrial carbon cycle can be studied using different approaches including (i) lab or field experiments (De Boeck et al., 2011; Beier et al., 2012; Song et al., 2019); (ii) observational data such as long-term forest observations (Anderegg et al., 2013) and local measurements of carbon exchange (Ciais et al., 2005; von Buttlar et al., 2018; Pastorello et al., 2020); (iii) indirect estimates from satellite observations (Ciais et al., 2005; Zhao and Running, 2010; Zscheischler et al., 2013; Stocker et al., 2019); and (iv) dynamical vegetation models (Ciais et al., 2005; Zscheischler et al., 2014a, b, c, d; Rammig et al., 2015; Xu et al., 2019; Bastos et al., 2020; Pan et al., 2020). Vegetation models offer the benefit of being able to analyse new hypotheses in a strictly controlled environment at global scale.

Despite considerable uncertainties in climate models, it is widely acknowledged that drought and heat extremes will increase in frequency and severity in many land regions in the future (Seneviratne et al., 2012). Though it is still uncertain exactly how these increases will affect the terrestrial biosphere, there are concerns they might substantially reduce the current terrestrial carbon sink (Reichstein et al., 2013). While coupled models of the land and atmosphere allow for a more complete representation of the feedback processes (Humphrey et al., 2021) than stand-alone land biosphere models, the analysis of results is more complicated for coupled models, since the coupling is different for different models and uncertainties depend not only on the land but also on the atmosphere module.

In this study, we aim to disentangle the differential effects of different frequencies of hot conditions, dry conditions, and compound hot-dry events on vegetation composition, carbon pools, and carbon dynamics. Our main motivation is to test the sensitivity of a commonly used vegetation model to differences in the climatology of the occurrence of hot and dry extremes and how these changes in drought and heat occurrence affect vegetation distribution and carbon dynamics. To this end, we force a dynamic global vegetation model, LPX-Bern v1.4, with six 100-year long climate scenarios featuring varying drought-heat signatures, i.e. different occurrence probabilities of dry events, hot events, and concurrent dry and hot events. These scenarios were sampled from 2000 years of present-day climate data from the EC-Earth climate model, as described in Sect. 2.1. They have a constant $CO_2$ concentration and do not contain long-term trends. The controlled environment of a model setup allows us to attribute changes in vegetation composition and carbon dynamics to differences in drought-heat occurrence.

## 2 Data and Methods

### 2.1 Forcing scenarios

Six forcing scenarios featuring different dry and hot signatures were used to run the vegetation model LPX-Bern. These scenarios, each 100 years long, were constructed from a large ensemble climate modelling experiment (Tschumi et al., 2021). 2000 years of simulated present-day climate data were created with the fully-coupled global climate model EC-Earth (v2.3, Hazeleger et al., 2012). The large ensemble was built out of 400 short five-year runs, which were unique in initial conditions and/or stochastic physics seed. EC-Earth combines atmospheric, oceanic, land, and sea-ice model components, and simulates the global climate including feedbacks between land and atmosphere. Within the ensemble the influence of forced climate change is small. We, therefore, assume all variability in the data set is due to natural variability in the climate system. While the global mean surface temperature in EC-Earth shows no significant bias, there can be biases at the regional and seasonal scale. In particular, there is a mean temperature difference of -0.5°C and a precipitation difference of 7 % over land, with regional biases being relatively large (up to -1.8°C in the tropics and 0.2°C in the extratropics, mostly in the very high latitudes). Many land regions show a wet bias in EC-Earth compared to CRU (43.5 % in the extratropics). A more detailed description of the biases can be found in Tschumi et al. (2021). The biases in the climate forcing compared to observational datasets implies that simulated vegetation cover based on this forcing may differ from observed vegetation cover.

The selection of the different scenarios from this data set was based on temperature and precipitation values during the time of the year where the vegetation is most active. Arguably, the vegetation is most vulnerable to climate extremes during the growing season. Therefore, for the scenario creation, we focused on the three months around the most productive month in the climatology. We identified the most productive month at each pixel, that is, the month with the highest climatological-mean net primary production (NPP) as simulated by LPX-Bern.

We selected the six different scenarios for each pixel separately based on mean temperature and precipitation over the three months around the month of highest NPP: *Control, Noextremes, Nocompound, Hot, Dry* and *Hotdry*. Years contributing to the scenarios were sampled based on quantiles of the three-month temperature and precipitation averages, where the quantiles were computed based on the full 2000-year EC-Earth output. If more than the required number of years fall into the quantiles in

question, a random selection was performed. If fewer years than necessary were available, some randomly chosen years were selected multiple times. For each of the *Hot*, *Dry*, and *Hotdry* scenarios, 50 years were sampled from the extreme quantiles and 50 years were randomly sampled from the rest. The reason for this is twofold. Firstly, for many pixels, not many years fall into the extreme quantiles. Sampling only 50 years from there reduces the number of times a year is re-sampled. Secondly, the mean climatology is kept more similar to the other scenarios if only half the years were sampled with extreme conditions and the other half from the rest.

This method of scenario creation, for each pixel separately, destroys any spatial coherence, so that the climate in a pixel is not correlated to the climate in nearby pixels. Furthermore, due to the sampling of individual years, there are always slight discontinuities between 31 December and 1 January in the climate forcing. The same is true for leap years since all leap days (29 February) were removed. We assume that these small discontinuities in the atmospheric forcing do not significantly affect our findings. The scenarios have a daily temporal and a $1° \times 1°$ spatial resolution. The scenarios were sampled from the percentiles of the EC-Earth data at each location separately as described in Tschumi et al. (2021) and summarized in Table 1.

**Table 1.** Sampling design for the six climate scenarios (see Tschumi et al., 2021)

| Scenario name | Sampling procedure |
|---|---|
| *Control* | 100 randomly selected years representing present-day climate |
| *Noextremes* | only years where temperature and precipitation lie between the $40^{th}$ and $60^{th}$ percentile |
| *Nocompound* | no years where both temperature and precipitation lie above the $85^{th}$ percentile or below the $15^{th}$ percentile |
| *Hot* | years where temperature exceeds the $85^{th}$ percentile and precipitation lies between the $40^{th}$ and $60^{th}$ percentiles |
| *Dry* | years where precipitation lies below the $15^{th}$ percentile and temperature lies between the $40^{th}$ and $60^{th}$ percentile |
| *Hotdry* | years where temperature lies above the $85^{th}$ percentile and precipitation lies below the $15^{th}$ percentile |

The scenarios differ little in their mean climatic conditions but strongly in the occurrence of dry events, hot events, and concurrent dry and hot events. More specifically, the difference in global mean temperature and precipitation between the scenarios is about 0.3 °C and 6 %, respectively. The *Hot* and *Hotdry* scenarios show an increase in heatwaves (based on cooling degree days, which is the sum of all exceedances over the $90^{th}$ percentile of the *Control* at each pixel) by up to 160 % compared to the *Control*. Dry event occurrences (based on the standardized precipitation index (SPI), which is used to identify severe meteorological droughts, defined as SPI <-1.5) are strongly increased for the *Dry* and *Hotdry* scenario, by up to 200 % compared to the *Control*. In the *Noextremes* and *Nocompound* scenarios, there is an overall decrease in dry events of up to -80 % and heatwaves up to -50 %. The pattern of concurrent dry and hot events is even more pronounced. There are no or very few concurrent dry and hot events in the *Noextremes* and the *Nocompound* scenario. Compound extremes are possible for the *Hot* and *Dry* scenario, but occur overall less often than in the *Control*. In the *Hotdry* scenario, however, concurrent dry and hot events occur up to 50 times more often than in the *Control*. A more in-depth description and analysis of these scenarios including the definition of dry and hot events are given in Tschumi et al. (2021).

## 2.2 LPX-Bern

LPX-Bern v1.4 (Lienert and Joos, 2018) is a Dynamic Global Vegetation Model based on the Lund-Potsdam-Jena (LPJ) model (Sitch et al., 2008). The model features coupled water, nitrogen, and carbon cycles and represents different types of vegetation using Plant Functional Types (PFTs). Here, only natural vegetation is considered, which is internally represented by eight tree PFTs and two herbaceous PFTs competing for resources and adhering to bioclimatic limits, which are listed in Table A1 as well as other process parameterizations (e.g. temperature dependence of photosynthesis or water balance). These bioclimatic limits and other parameters as well as process representation can differ from model to model, leading to a different response of the vegetation to extreme climatic events. In LPX-Bern, tree coverage is restricted to 95% of the grid cell. If the total fraction summed over all PFTs exceeds 1, the plants that were the least productive are killed, representing self-thinning. Mortality can also occur if a PFT's bioclimatic limits are reached due to heat stress, negative NPP, or depressed growth efficiency (Sitch et al., 2003). As an example, the bioclimatic parameter governing the upper limit of temperature is implemented in LPX-Bern by inducing mortality proportional to the number of days in the year where this threshold is exceeded. Other models may not only use different values for the threshold and a different relationship between mortality and exceedance, but an altogether different parameterization. This will in turn influence the response to the heat stress in the model.

In this study, daily temperature, precipitation, and incoming short-wave radiation are provided to the model. Additionally, the model uses information on the soil type (Wieder et al., 2014), $CO_2$ concentration in the atmosphere at 2011-level (389.78 ppm), and nitrogen deposition, also at 2011-level (Tian et al., 2018). Each scenario simulation was preceded by a 1500 year long spin-up, which was forced with climate data of the same scenario ('individual spin-up'). To test how fast vegetation composition and net ecosystem exchange reach a new equilibrium under an altered frequency of dry and hot events, we also performed simulations in which the spin-up was based on climate from the *Control* ('shared spin-up'). By running the model with two different spin-ups per scenario, we explore the model equilibrium and how fast the model reacts after a step change in the frequency of extreme events.

LPX-Bern represents natural vegetation with ten PFTs, as described above. For the following analysis, we aggregate them into four broader classes, namely Tropical trees (including tropical broad-leafed evergreen and tropical broad-leafed raingreen trees), Temperate trees (including temperate needle-leafed evergreen, temperate broad-leafed evergreen and temperate broad-leafed summergreen trees), Boreal trees (including boreal needle-leafed evergreen, boreal needle-leafed summergreen and boreal broad-leafed summergreen trees), and Grasses (including temperate and tropical herbaceous). The dominant vegetation class in the control simulation for each pixel, including its fractional cover (the fraction of a grid cell covered with a certain vegetation class), is shown in Fig. 1. Pixels where the total fractional coverage is smaller than 0.1, corresponding to desert regions, are masked white.

## 3   Results

We report how different stationary climate conditions (i.e. without long-term trends) with varying intensities of dry events, hot events and compound dry-hot events affect vegetation coverage (Sect. 3.1) as well as carbon pools and carbon fluxes (Sect. 3.2).

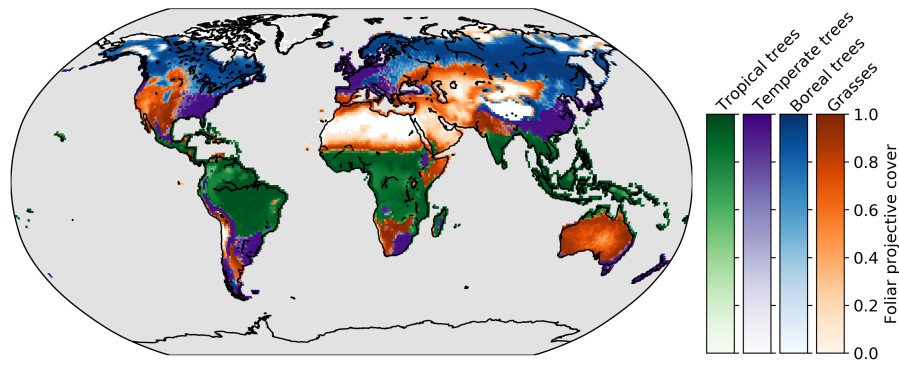

**Figure 1.** Dominant vegetation class (mean over time) in the *Control* simulation. The intensity (color bars) shows the fractional coverage of each dominant class.

These results are based on the simulations using the individual spin-up. In Section 3.3 we report how quickly LPX-Bern reaches a new equilibrium by running simulations for each scenario that use the climate of the *Control* scenario during spin-up (shared spin-up).

### 3.1 Changes in vegetation coverage and associated NPP changes

The different dry and hot scenarios lead to a change in fractional vegetation coverage (Fig. 2a). Trees generally benefit from a climate with no dry and hot events. The increase in tree cover is stronger for higher latitudes. While the relative difference in global mean Tropical tree cover is 1.2 %, it is 9.4 % for Boreal trees for the *Noextremes* scenario (green bars in Fig. 2a). Regionally, this increase can be much larger. Total tree cover for the mid-west of the U.S., for example, is increased by up to 400 % and there is a similarly large increase in South Africa (results not shown). These are regions with nearly no trees in the *Control* scenario (Fig. 1). A smaller, but still large increase of up to 100 % is observed in South America, southern Africa and large parts of Eurasia. Grass coverage in turn decreases to make room for the trees. To a lesser extent, the same pattern also holds for a climate with no compound extremes, which however does feature univariate extremes (blue bars in Fig. 2a). The increase of tree coverage towards higher latitudes is also evident for the *Hot* scenario, while for this scenario grass cover does not change compared to the *Control* (red bars in Fig. 2a). The *Dry* and, even more strongly, the *Hotdry* scenarios lead to an overall decrease of tree coverage (orange and purple bars in Fig. 2a, respectively). The decrease is particularly strong for Temperate tree coverage in the *Hotdry* scenario (-5.6 %), while there is little change in Boreal tree cover. At the regional scale, the decrease is largest in the mid-west of the U.S. with up to -80 % as well as up to -50 % in mid-Eurasia. For the *Hotdry* scenario, the overall decrease in tree cover is compensated by an increase in grass cover, mainly in the U.S., Europe, mid-Eurasia and southern South America, in contrast to the *Dry* scenario, in which grass cover also decreases. While it is generally true that grasses seem to compensate for declining tree coverage, the compensation is not necessarily complete. As an effect, the total sum of fractional plant cover may change as well. However, at the global scale, there is hardly any change in

fractional coverage between the scenarios (not shown). Overall, the differences in vegetation cover between the scenarios are smallest for Tropical trees and tend to be similarly ordered, but larger in magnitude, for the other vegetation classes.

The above-described relative differences in coverage directly translate into changes in annual NPP (Fig. 2b). In particular, if tree or grass coverage increases, so does NPP and if coverage decreases, we find an associated decrease in NPP. Overall, at the global scale, the variability in the relative differences in NPP is larger than the variability in the relative differences in vegetation cover (compare lengths of bars in Fig. 2a to Fig. 2b).

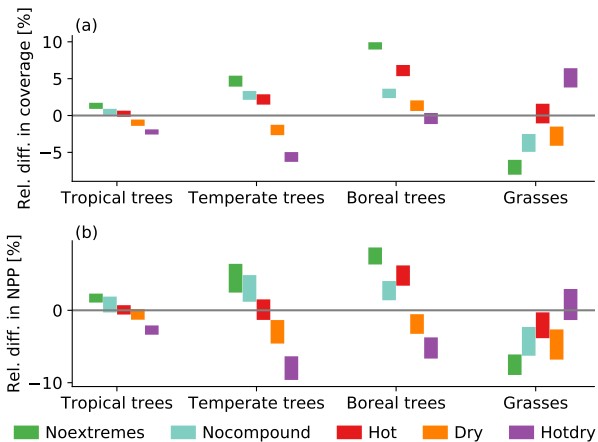

**Figure 2.** Relative difference of the scenarios to the *Control* for (a) coverage and (b) annual NPP. The bars show the minimum to maximum range over the 100-year long simulations.

We compare the spatial patterns of the differences of tree (all tree types aggregated) and grass cover between the two scenarios with the strongest effect and the *Control*, i.e. *Noextremes-Control* and *Hotdry-Control*, in Fig. 3. In the *Noextremes* scenario, tree cover increases on all land pixels compared to the *Control*, especially in western North America and Mid-Eurasia (Fig. 3a). In contrast, grass cover decreases everywhere except in very dry regions such as the Sahara, the Arabian Peninsula, and Australia, where a constant climate without extremes leads to a slight increase in grass cover (Fig. 3b). For *Hotdry*, tree cover decreases in most regions except the very high latitudes, compared to the *Control* (Fig. 3c), while grass coverage increases except for very dry regions (Fig. 3d).

## 3.2 Changes in carbon dynamics

The effects of the scenarios on vegetation coverage (Sect. 3.1) are reflected by the globally aggregated carbon fluxes and pools (Fig. 4). The response of NPP to the replacement of trees with grasses and vice versa is varied, as it strongly depends on environmental conditions and vegetation composition. Generally, NPP is greater for trees than for grasses, which implies that global NPP is larger in a world with more trees and smaller if more forest area is replaced by grassland. Consequently, *Noextremes*, *Nocompound*, and *Hot* generally show higher or similar flux magnitudes compared to the *Control*, whereas fluxes

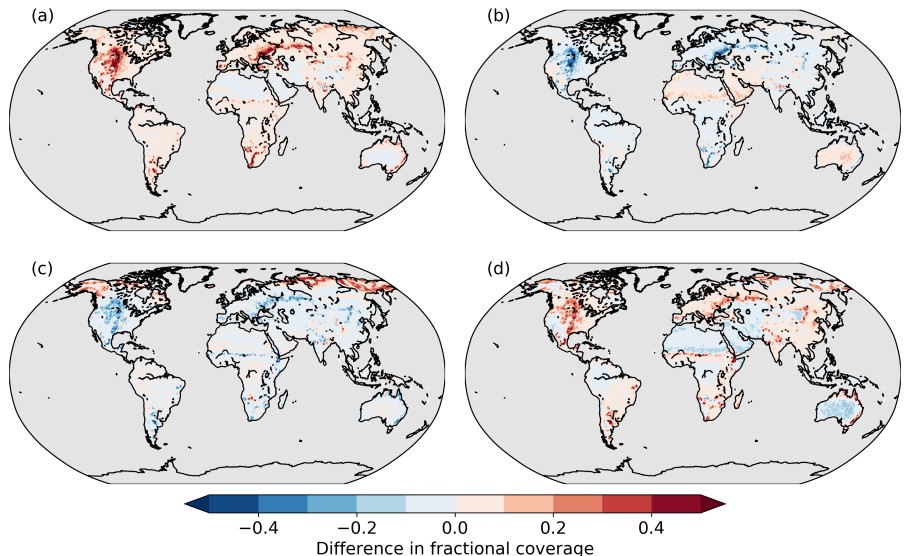

**Figure 3.** Difference in fractional coverage of (a) *Noextremes* trees, (b) *Noextremes* grasses, (c) *Hotdry* trees and (d) *Hotdry* grasses compared to the *Control*.

are strongly decreased for *Dry* and *Hotdry*, by up to more than -4 % for global gross primary production (GPP) in *Hotdry* (Fig. 4a). Interestingly, although grass cover is increased in the *Hot* scenario (Fig. 2a), NPP in grasslands is reduced (Fig. 2b), explaining the lack of change in global NPP for the *Hot* scenario (Fig. 4a). Relative carbon flux reductions can be very large for some regions, for example up to -80 % in the mid-west of the U.S., mirroring the decrease in tree cover. Similar patterns are evident for changes in global vegetation carbon (Fig. 4b). Overall, relative differences are much smaller for global soil carbon.

We further explore the spatial patterns in the differences of NPP separately for trees (all tree types aggregated) and grasses between the two scenarios with the strongest effect, i.e. by looking at *Noextremes-Control* and *Hotdry-Control* (Fig. 5). NPP of trees increases nearly everywhere in *Noextremes* compared to the *Control*, by up to 200 gC m$^2$ yr$^{-1}$ in some regions in the mid-west of the U.S. (Fig. 5a). NPP of grasses shows slight increases in the lower latitudes but strong decreases in the higher latitudes, which are of similar magnitude as the increases in tree NPP (Fig. 5b). The pattern is more diverse for *Hotdry*, where NPP of trees generally decreases in the low-to-mid latitudes by up to -150 gC m$^2$ yr$^{-1}$ but increases in the very high latitudes (Fig. 5c). NPP of grasses tends to increase in most regions except some very dry regions in the Sahara and Middle East, Australia, Namibia, and the Southwest of the U.S. (Fig. 5d).

Finally we investigate whether the interannual variability in NPP for four vegetation classes changes between the *Control* and the different scenarios. Overall, interannual variability in NPP is smallest in Tropical and Temperate trees and largest in Boreal trees (Fig. 6). Most scenarios tend to decrease variability in particular for trees, with *Noextremes* leading to significant decreases in all vegetation classes. In contrast, *Hotdry* tends to increase variability, though the difference to the *Control* is only

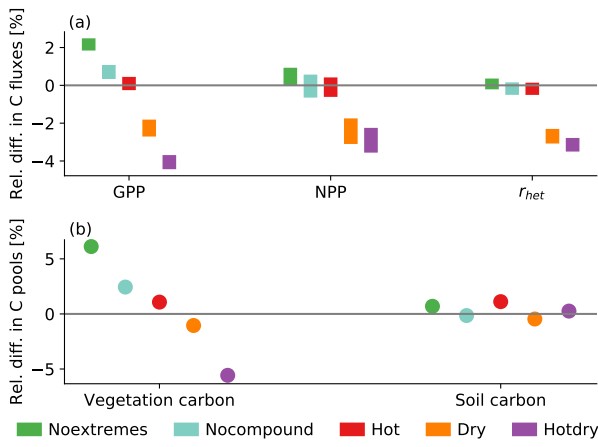

**Figure 4.** Relative difference of the scenarios to the *Control* for (a) the global annual GPP, NPP, and heterotrophic respiration ($r_{het}$) as well as (b) vegetation carbon and soil carbon. The bars in (a) show the minimum to maximum range of the 100 year-long simulations. Because the interannual range for carbon pools in (b) is very small we only show the mean over the 100 years.

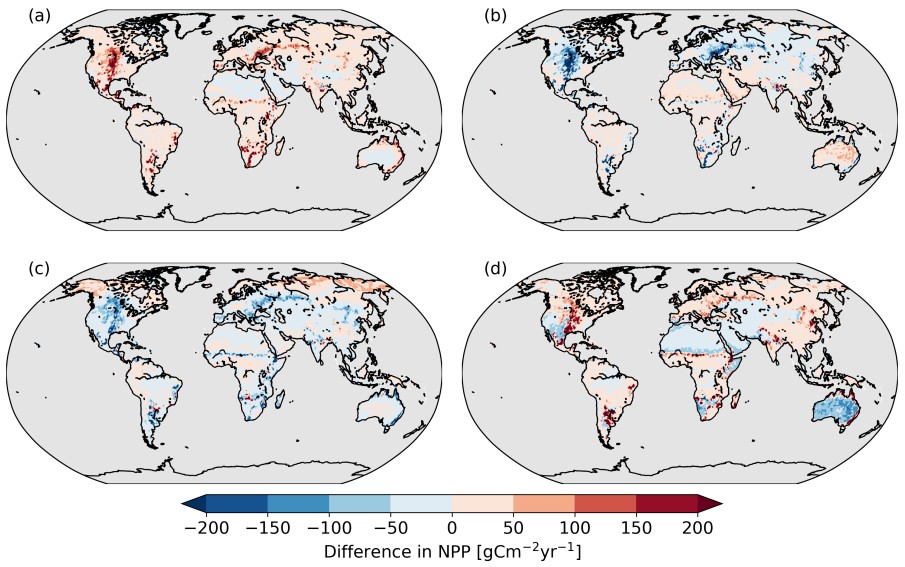

**Figure 5.** Difference in NPP for (a) *Noextremes* trees, (b) *Noextremes* grasses, (c) *Hotdry* trees and (d) *Hotdry* grasses compared to the *Control*.

significant for Boreal trees and Grasses. For Grasses, also the *Hot* and the *Dry* scenario lead to a significant increase in NPP variability.

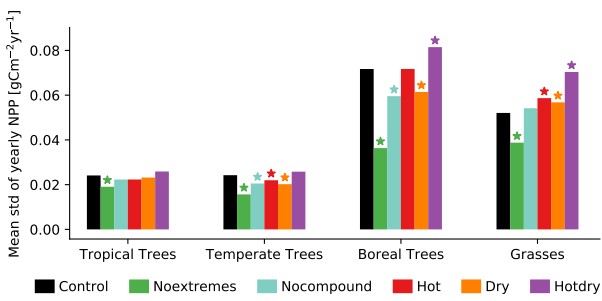

**Figure 6.** Variability of NPP (calculated as interannual standard deviation across years for the four vegetation classes with the mean taken over all grid cells). The stars show the scenarios that are significantly different from the *Control* at a 5 % significance level (based on a t-test).

## 3.3 Path to model equilibrium

We explore how fast vegetation composition and net ecosystem production adjust towards a new equilibrium after a step-like change in extreme statistics, in this case a change in the frequency of hot and/or dry extremes. To this end, we analyse the 100-yr scenario simulations that started from the shared model spin-up forced by the *Control* climate. At the start of each scenario simulation, frequencies of dry and hot events suddenly change from those in the *Control* climate to those in the scenario.

Using the simulations based on the shared spin-up, we explore whether LPX-Bern reaches a new equilibrium (measured in terms of stable vegetation composition and neutral net ecosystem production) within the 100-year simulations after frequencies of dry and hot events suddenly change from *Control* to the different scenarios. Overall, the *Noextremes* and the *Hotdry* scenarios cause the largest disturbance in vegetation cover (Fig. 7). For most vegetation classes and most scenarios, the scenario simulations starting from the shared spin-up are within the range of variability of the scenario simulation starting from an individual spin-up at the end of the simulation. Exceptions are Tropical trees in the *Noextremes* and the *Hot* scenarios, Temperate trees in the *Hot* and the *Hotdry* scenarios and Grasses in the *Hotdry* scenario. The strongest response in vegetation cover occurs in the first 20 years. Grasses show a particularly fast response in the *Hotdry* scenario, where there is an initial decrease in coverage followed by a rapid increase. The reason for this seems to be that (predominantly temperate) grasses that are adapted to the climate in the control quickly die due to the frequent hot and dry conditions but then a regrowth of (predominantly tropical) grasses that can tolerate such conditions occurs. Overall, the above results suggests that, for the more extreme scenarios, 100 years may not be enough to fully reach equilibrium after a sudden change in dry and hot event occurrences.

The findings based on vegetation cover are confirmed when investigating the temporal evolution of global annual net ecosystem production (NEP) in the simulations with shared spin-up (Fig. 8). Again, the disturbance is largest for the *Noextremes* (about 1 PgC yr$^{-1}$ more uptake at the beginning of the simulation) and the *Hotdry* scenario (about 3 PgC yr$^{-1}$ less uptake at the beginning). In all scenarios, global annual NEP converges towards 0 at the end of the 100-year simulations and varies within the range of interannual variability of the individual spin-up simulations. Nevertheless, NEP is slightly larger than 0 in

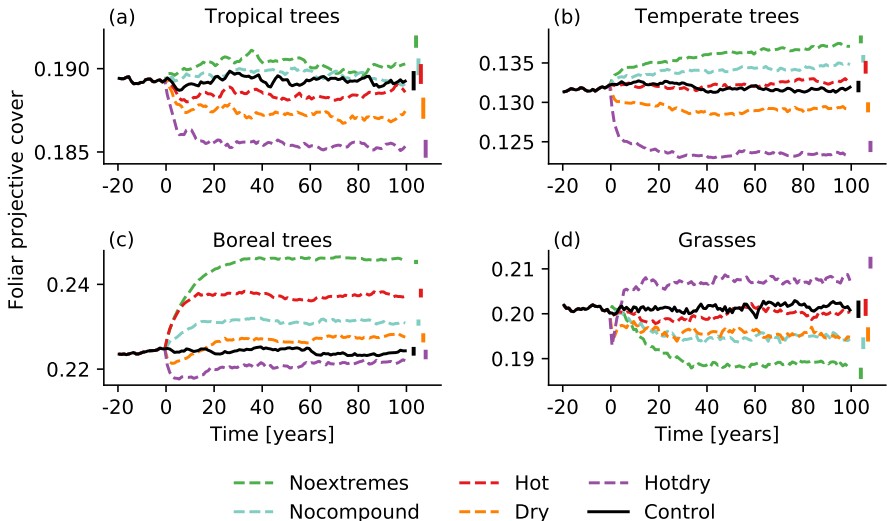

**Figure 7.** Time series of the fractional coverage (foliar projective cover) from the simulations that use the shared spin-up (black line). Scenarios are shown in colored dashed lines for (a) Tropical trees, (b) Temperate trees, (c) Boreal trees, and (d) Grasses. The first 20 years (-20 to 0) represent the last 20 years of the shared spin-up. The variability (minimum to maximum) in vegetation cover in the individual spin-up simulation (spin-up uses data from the respective scenarios) is indicated by the bars on the right-hand side. Note the different ranges of the y-axes.

the *Noextremes* scenario and slightly smaller than 0 in the *Hotdry* scenario even at the end of the simulation, indicating that not all carbon pools are in full equilibrium after 100 years.

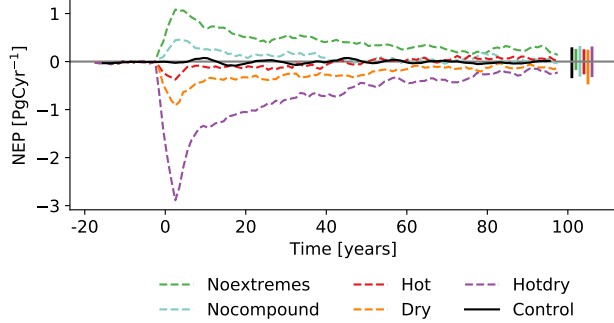

**Figure 8.** Time series of global annual NEP from the simulations that use the shared spin-up (black line). Scenarios are shown in colored dashed lines. The first 20 years (-20 to 0) represent the last 20 years of the shared spin-up. The variability (minimum to maximum) in global NEP in the individual spin-up simulation (spin-up uses data from the respective scenarios) is indicated by the bars on the right hand side. A 5-year moving average was applied to smooth the time series.

## 4 Discussion

Using stationary climate scenarios with varying drought-heat signatures and a dynamic vegetation model we show that different occurrence frequencies of dry, hot, and compound dry-hot events lead to differences in vegetation coverage and related differences in global NPP (Fig. 2). The fraction of land area covered with vegetation is similar in all scenarios. However, there are shifts in coverage and NPP between vegetation classes. A key finding is that the climate, as represented by the *Noextremes* scenario, which features no extreme droughts or heatwaves and relatively little interannual variability, favors tree coverage (Fig. 2). This is evident in the tropical biomes to some extent but even more evident at higher latitudes. For trees to grow well, typically more stable environmental conditions are needed as compared to grasses (Sitch et al., 2003). For example, the biomass of grasses, with their fast biomass turnover and short life cycle, recovers much faster after an increase in mortality, e.g., due to a drought-heat event, than tree biomass.

Hence, overall, a more stable climate with few extremes is very beneficial for trees. In models such as LPX-Bern, trees are favored over grasses. In particular, they get priority for foliar coverage if conditions are suitable for tree growth. This explains why, in a more stable (i.e., less variable) climate, tree cover increases and grass cover decreases, and vice versa.

While a climate with more heatwaves has little influence on tree coverage in the tropics, it tends to increase coverage in higher latitudes (Fig. 2). Trees in higher latitudes are typically temperature limited (Way and Oren, 2010). So a climate with more heatwaves alleviates some of these temperature constraints. While overall more heatwaves increase tree coverage globally, there are strong regional variations, meaning that not everywhere higher temperatures lead to more growth (Ruiz-Pérez and Vico, 2020). In higher latitudes, more frequent heatwaves mean overall warmer temperatures during the growing season without necessarily exceeding the temperature limit of boreal trees, while in other regions such a limit might be reached more quickly, leading to a decrease in tree cover. Grass coverage does not significantly change for the *Hot* scenario compared to the *Control*.

If water is restricted, as it is for the *Dry* scenario, tree coverage is slightly reduced overall. However, unlike in the other scenarios, grasses in a dry climate do not compensate for changes in tree coverage. Rather, grass coverage is decreased as well. This likely happens because grasses tend to grow in already dry regions, where tree coverage is unlikely. If these regions get drier, it might even get too dry for grasses to grow. When comparing the *Hot* and *Dry* scenarios, we see that the effects on global NPP as well as the vegetation carbon pool are more negative for the *Dry* than *Hot* scenario (Fig. 4). A drought event, therefore, does not have to be as extreme as a heat event to have a comparable impact, which is also supported by findings of Ribeiro et al. (2020).

The scenario with frequent compound hot and dry extremes clearly causes the strongest response and leads to a reduction in tree coverage across all climate zones. Hence, here even the warmer conditions in the northern latitudes that generally promote tree growth are superseded by the negative impacts of droughts (Belyazid and Giuliana, 2019; Ruiz-Pérez and Vico, 2020), though the effect is less pronounced for Boreal trees than for Temperate trees. Grass coverage, on the other hand, increases because it can fill the areas that were previously covered by trees. In dry regions, however, grass coverage is reduced for the *Hotdry* scenario as well, likely because here likely dryness thresholds under which vegetation cannot grow anymore are

frequently exceeded. Global NPP as well as vegetation coverage is overall reduced for this scenario compared to the *Control* (Fig. 4).

Generally, trees grow nearly everywhere if the climate is favourable and features few extremes, leading to a reduction in grass cover. Only in dry regions do we observe an increase in grass coverage. There, conditions might still be unfavourable for
trees to grow, but grasses benefit from the stable climate. In contrast, in a climate with frequent droughts and heatwaves, tree coverage is generally reduced, leaving room for grasses to grow, except in already dry regions, which become too dry even for grasses.

Globally, the effects of extremes are larger in the extratropics than they are in the tropics. The effects on Tropical trees are small for all scenarios compared to the *Control*, including the *Hot* and *Dry* extremes scenarios. One reason for this might be that
strong evaporative cooling is maintained in tropical forests, even in a drier climate (Bonan, 2008) since the tropics (in particular tropical forest) are not so much water-limited but rather energy limited. However, case studies on recent droughts in the Amazon forest show how tropical forests can be negatively affected by drought conditions (Doughty et al., 2015; Feldpausch et al., 2016; Machado-Silva et al., 2021). The variability between the scenarios is small for Tropical trees and larger for Temperate and Boreal trees. The latter biomes are more water- and/or temperature-limited than the tropics and therefore react more strongly to
variations in these variables. Grasses also show quite a large variability between scenarios owing to the fact that grasses react quicker to climate variations, meaning they die and regrow faster than trees (Ahlström et al., 2015).

While vegetation carbon displays a pattern that correlates with the changes in coverage, the same is not true for soil carbon. Rather, the changes in soil carbon (Fig. 9) resemble the changes in grasses (Fig. 3).

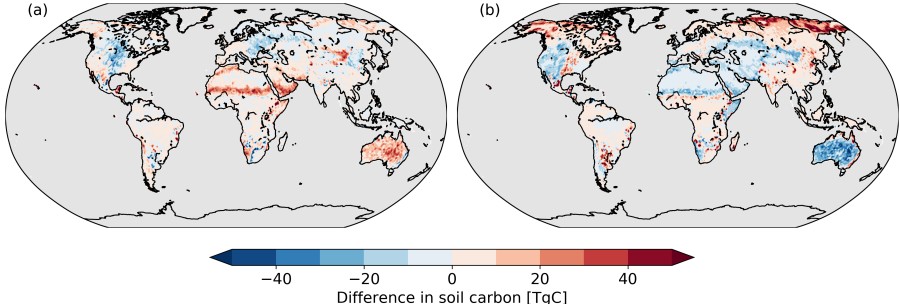

**Figure 9.** Difference in soil carbon for (a) *Noextremes* vegetation (b) and *Hotdry* vegetation to the *Control*.

Choosing an appropriate spin-up when modelling vegetation and the carbon cycle is important to make sure the model is
285 in equilibrium. In our case, 1500 years seems appropriate, since the constant runs are stable over the 100 years. Starting with the same spin-up (based on the *Control* scenario) and a step change in extreme event occurrence, most but not all scenarios converge to the equilibrium that is reached when doing the spin-up with the scenario forcing within 100 years (Fig. 7). Given the trajectories, we do not expect the runs with shared spin-up to reach the same end point as the runs with individual spin-up, even if the simulations were prolonged. Other vegetation models might have other response times to such a step change in

extreme event characteristics. For the main analysis, we used the 'individual spin-up' runs, since these are the runs where the model had time to reach full equilibrium.

Scenarios, where the occurrence of heatwaves, droughts, and drought-heat events is changed in a step-like manner, reveal the characteristic time scales and magnitudes of the adjustment of a system, here the land biosphere, to the change. Our simulations reveal that plant coverage and NPP adjust on decadal time scales (Fig. 7) to altered extreme event statistics, while, in addition, multi-decadal-to-century response time scales are evident for global NEP (Fig. 8). The response time scales and magnitudes of change are likely model specific to some extent. It would be illustrative to probe the response to step changes using other models. Though the setup of the step change in the occurrence of droughts and heatwaves is somewhat unrealistic, long-term trends in the dependence between temperature and precipitation have been detected in climate model projections (Zscheischler and Seneviratne, 2017). Such changes in the dependence structure can be quite relevant, for instance they may exacerbate climate change impacts on crops (Lesk et al., 2021).

We run the vegetation model offline, that is, there is no feedback from the land surface to the climate, and keeping the atmospheric $CO_2$ level constant. Processes in the real world might be more complex. Especially $CO_2$ fertilization, where higher $CO_2$ concentrations lead to a more efficient uptake of $CO_2$ by the plants and thus less chance to lose water through open stomata, may modulate how hot and dry conditions affect vegetation and carbon dynamics in the future (Domec et al., 2017; De Kauwe et al., 2021).

All results, such as the exact changes in vegetation distribution and carbon uptake, are somewhat sensitive to the choice of the dynamic global vegetation model and the employed climate model. Every model has biases and limitations which could be discussed at length, but for argument's sake we will only discuss some of them briefly. One important component in LPX are the bioclimatic limits, as already mentioned in Sect. 2.2. Mortality induced by maximum temperature only affects tropical trees. One could imagine a different extreme response if this parameter also applied for grasses. As it is, C4 grasses are very water-efficient in LPX, which leads to Australia being a bit too green in our simulations compared to observations, as an example. This could also explain why grasses thrive in the *Hotdry* scenario. A potential increase in atmospheric $CO_2$ conditions as it is predicted by socio-economic scenarios would further alleviate drought stress and thus benefit C4 grasses. The parameterization of the water balance is another possible factor that greatly influences the response to dry conditions. LPX has a relatively simple supply and demand driven water limitation and for instance does not consider effects of Xylem damage (Arend et al., 2021). Overall, models may differ strongly depending on model parameterizations and process representations (Paschalis et al., 2020). Furthermore, some uncertainties also arise from the model setup. For example, land-atmosphere feedbacks may play an important role (Humphrey et al., 2021), which are not considered in such an offline model setup as we have conducted in this study. Considering the number of uncertainties that may govern vegetation and carbon cycle response to varying drought-heat signatures, a model intercomparison project using our scenarios as forcings for different vegetation models has already been set up and may reveal insights on how model differences affect the results.

## 5 Conclusions

It is widely acknowledged that extreme climate events can have large impacts on ecosystems and society. This study investigates the effects of different drought-heat occurrences in six hypothetical climate scenarios on vegetation distribution and terrestrial carbon dynamics, as simulated by the LPX-Bern dynamic global vegetation model. Generally, effects of changes in extreme event frequency are more pronounced in the extratropics than in the tropics. We found that global carbon cycle variability is most stable in a climate without any extreme events, which favours more tree cover and a higher global terrestrial carbon stock. The effects on vegetation cover and carbon stocks and fluxes of a climate with many heatwaves are generally smaller than the effects of a climate with many droughts. The largest effect, however, has a climate with frequent concurrent droughts and heatwaves. Here, forest cover and global vegetation carbon is strongly reduced. Grasses, in contrast, are more abundant. These effects surpass the simple linear combination of the effects of single droughts and single heatwaves.

Overall, our results highlight the importance of considering compound events when analysing impacts of climate extremes. Impacts may potentially be underestimated when only looking at single event extremes instead of compounding extremes. Furthermore, the results suggest that uncertainties in projections of vegetation distribution and carbon dynamics in Earth system models may stem from different drought-heat signatures in the atmospheric module (Zscheischler and Seneviratne, 2017), in addition to structural model differences in the vegetation component. It is important to investigate and understand these issues in order to improve models as well as our knowledge about extreme events and their processes, which may lead to significant impacts on society and ecosystems.

*Data availability.* The forcing scenarios are described in (Tschumi et al., 2021) and can be accessed via zenodo (10.5281/zenodo.4385445). The LPX-Bern simulations are very large and are available from Elisabeth Tschumi (elisabeth.tschumi@unibe.ch).

## Appendix A

*Author contributions.* J.Z. conceived the study. E.T. performed all analysis, performed model simulations with LPX-Bern, created all figures and wrote the first draft. K.v.d.W. performed the model simulations with EC-Earth. S.L. and F.J. helped with the setup of LPX-Bern and interpretation of the results. All authors contributed substantially to the writing and revisions of the manuscript.

*Competing interests.* The authors declare that they have no competing interests.

**Table A1.** Bioclimatic limits of the ten available plant functional types in LPX-Bern.

| | Minimum coldest monthly mean temperature | Maximum coldest monthly mean temperature | Minimum growing degree days (at or above 5°C) | Upper limit of temperature |
|---|---|---|---|---|
| TrBE (Tropical Broadleaf Evergreen) | 15.5 | no limit | 0 | no limit |
| TrBR (Tropical Broadleaf Raingreen) | 15.5 | no limit | 0 | no limit |
| TeNE (Temperate Needleleaf Evergreen) | -2 | 22 | 900 | no limit |
| TeBE (Temperate Broadleaf Evergreen) | 3 | 18.8 | 1200 | no limit |
| TeBS (Temperate Broadleaf Summergreen) | -17 | 15.5 | 1200 | no limit |
| BoNE (Boreal Needleleaf Evergreen) | -32 | -2 | 550 | 30 |
| BoNS (Boreal Needleleaf Summergreen) | no limit | -2 | 350 | 30 |
| BoS (Boreal Broadleaf Summergreen) | no limit | -2 | 550 | 30 |
| TeH (Temperate Herbaceous) | no limit | no limit | 0 | no limit |
| TrH (Tropical Herbaceous) | no limit | no limit | 100 | no limit |

*Acknowledgements.* The authors acknowledge the European COST Action DAMOCLES (CA17109). E.T. and J.Z. acknowledge the Swiss National Science Foundation (Ambizione grant 179876). J.Z. further acknowledges the Helmholtz Initiative and Networking Fund (Young Investigator Group COMPOUNDX, Grant Agreement VH-NG-1537). K.v.d.W. acknowledges funding for the HiWAVES3 project from the National Natural Science Foundation of China (41661144006), funding was supplied through JPI Climate and the Belmont Forum (NWO ALWCL.2 016.2 and NSFC 41661144006). F.J. and S.L. acknowledge funding from the Swiss National Science Foundation (Grant no. 200511) and from the European Union's Horizon 2020 research and innovation programme under grant agreement No 821003 (project 4C, Climate-Carbon Interactions in the Current Century). The work reflects only the authors' view; the European Commission and their executive agency are not responsible for any use that may be made of the information the work contains.

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
