# Peer review of "The effects of varying drought-heat signatures on terrestrial carbon dynamics and vegetation composition"

_Biogeosciences, 2021_

## Author Comment (AC1)

**We thank Martin de Kauwe for his constructive comments. A point-by-point response follows below.**

Tschumi et al. explore the timely topic of the impact of combined drought/heat extremes on terrestrial carbon dynamics, using the LPX model. The authors construct a suite of potential forcings and use a factorial approach to examine the follow-through impact on carbon/tree dynamics. I found the paper well written, the methods are well constructed and the analysis broadly interesting. I did feel that more could have been done to explain key assumptions (see below) and in particular to leverage these assumptions in the discussion text when discussing impacts/broadening how transferable the results are across models. I think the equilibrium analysis needs more motivation in the paper, currently, 2 of the 9 figures relate to this and there is no mention in the abstract.

**Thank you for your helpful comments and overall positive feedback.**

The first major issue relates to what LPX assumes, particularly with respect to temperature. The manuscript lacks information on the assumed bioclimatic limits and that photosynthesis and respiration estimates are simply down-regulated if temperature bounds are exceeded (I'm going off what I know about LPJ-GUESS, so please correct me). I think this information should be in the methods (a table by PFT) as it is likely to impact on how transferable the results are across models. For example, how much of the impact of the hot (or hotdry) is due to the assumed upper bounds of the model? Are these sensible in all geographic locations (see Australia, for example, where I also note the model has a continent of grasses - https://bg.copernicus.org/articles/18/2181/2021/). I would anticipate some return to this in the discussion when the authors try to contextualise their results. I want to be clear though, I'm not suggesting this invalidates anything the authors have done, but I would nevertheless anticipate a degree (a lot) of sensitivity to the assumptions of the DGVM. Similarly, some text on how well the model simulates the impact of drought.

**We will include a table with LPX's specific bioclimatic limits in the appendix (similar to the table shown here). In addition, we will extend the Methods section on model processes and discuss some of the models limitations in the Discussion section.**

|  | minimum coldest monthly mean temperature | maximum coldest monthly mean temperature | minimum growing degree days (at or above 5°C) | upper limit of temperature of the warmest month |
|---|---|---|---|---|
| TrBE | 15.5 | no limit | 0 | no limit |
| TrBR | 15.5 | no limit | 0 | no limit |
| TeNE | -2 | 22 | 900 | no limit |
| TeBE | 3 | 18.8 | 1200 | no limit |

| | | | | |
|------|----------|----------|------|----------|
| TeBS | -17 | 15.5 | 1200 | no limit |
| BoNE | -32 | -2 | 550 | 30 |
| BoNS | no limit | -2 | 350 | 30 |
| BoS | no limit | -2 | 550 | 30 |
| TeH | no limit | no limit | 0 | no limit |
| TrH | no limit | no limit | 100 | no limit |

This point about assumptions can be extended when the authors discuss how competition changes fractional cover with scenarios. I think the C4 grass component of the model will have assumed bioclimatic limits, in some places (Australia I know for certain), these are likely to bias interpretations …

**The Discussion will be extended with some limitations and biases, which include a wet bias in Australia in the EC-Earth forcing data and the fact that our C4 grasses are very water-efficient in LPX, which both leads to Australia being a bit too green in our simulations compared to observations.**

The second major issue relates to missing insight into how realistic the output from EC-Earth is? Is it biased warm/cold, wet/dry. The model will clearly have biases and I'm not suggesting that these biases would warrant the paper not being accepted; however, they do affect the subsequent analysis through LPX. It is really important that some text is added about this I feel. Having personally been recently looking at the mean and variability bias in future precipitation data over Australia, I think this is a significant issue.

**Note that we discussed the biases in EC-Earth in detail in an earlier manuscript: https://rmets.onlinelibrary.wiley.com/doi/full/10.1002/gdj3.129. For completeness, we will extend the section on the EC-Earth bias in the Data section and cross reference to our earlier paper. It is important to keep in mind that the analysis was done by looking at changes relative to the control (which has the same bias as the rest of the scenarios) and not to observations, therefore the EC-Earth bias does not play a relevant role when analysing the differences between scenarios.**

Finally, I think it would be great to add some figure (timeseries) that shows a case study of one of these very extreme summers to see how the model is behaving, for some region. I might be alone here, but I'd appreciate seeing how the model dynamics in terms of uptake/respiration play out across an extreme scenario vs one without events. It is quite hard to get this message from the maps.

**Thank you for this suggestion. We have tried to come up with a time series illustration that would be useful for the overall story but did not succeed. Generally we look at how changes in the climatology of the occurrence of droughts and heatwaves affect vegetation distribution and carbon dynamics.**

Hence focusing on single events would not be very informative but rather illustrate the generic response of LPX to certain types of climate extremes. To address the reviewer's comment we will expand the Introduction to make sure the scope of this study is clear and add some discussion on how LPX deals with droughts and heatwaves.

Minor
* * *
- Line 23: Allen et al. isn't an appropriate citation for CO2-induced changes in WUE.

**We will change this reference to De Kauwe et al. (2013) (Global Change Biology, 19, 1759-1779, https://doi.org/10.1111/gcb.12164).**

- I think the scenarios sound well thought through and I note what I suspect is a deliberate naming choice to call one of the "dry" rather than drought.  I think this is the right choice as the authors can't be sure it indeed does sample a drought (unless they check). However, the text surrounding Table 1 talks in terms of "drought". I think it would be better to be consistent with the language of the scenarios (?). I note further on they talk about checking for drought using SPI. Could they perhaps add a few sentences to explain a bit more about how they did this, it is currently unclear for me.

**We will make sure that we always talk about 'dry' conditions instead of 'drought'. We will expand on the SPI analysis and cross-reference our previous paper where this is explained in more detail.**

- I thought the equilibrium text was interesting but I'm not sure that the context was really given? If I missed it, I apologise but I would have liked to have seen more in the intro/methods to explain motivation. I also wonder whether each change in Fig 7 would be better on the same y-axis so the reader can compare relative impacts.

**The equilibrium analysis provides some insight into how quickly the model responds to a sudden change in forcing, here a change in frequency of hot and dry events, and how quickly the model converges to a new equilibrium. We will add some explanation to motivate this aspect better. Regarding Figure 7, we prefer to keep the different y-axes because the results are easier to read this way. See the following version of Figure 7 which has the same y-axes.**

[Figure]

- With the maps, consider chopping off Antarctica to show more of the land surface.

**We will consider cutting off Antarctica from the maps.**

Martin De Kauwe

---

## Author Response (AR1)

**We thank the reviewers and the editor for their time in reviewing our manuscript and for their constructive comments. Below we provide a point-by-point response to each of the comments.**

Associate editor decision

Comments to the author:
The reply to the reviewers gives a good direction of the new version. Christopher Reyer suggests to be more explicit on the response mechanisms in the vegetation model, and Martin de Kauwe asks for time series that illustrate the response of the model. I think this could be combined by explaining (textually, graphically) which set of (photosynthesis?) processes in the LPX model cause the effect of compounding extremes to be larger than the sum of the two separate extremes. Apparently there is a representation of conditional sensitivity to one driver that depends on the state of the other driver. Please explain where this happens in the model.

**We assume the editor refers to the statement in the abstract: "...the effects from the *Hotdry* scenario are stronger than the effects from the *Hot* and *Dry* scenario combined." This statement is a synthesis out of figs. 2 and 4 primarily, where we show the mean response of the model to different frequencies of hot and dry events. This is a somewhat emergent response of many effects and is difficult to narrow down to individual processes. We have, however, added much more information on the model's internal structure to better motivate and explain why LPX shows the observed responses.**

I've labeled the new submission as a "major revision", but this is motivated by the fact that both reviewers expressed to be willing to see the manuscript again. I think the revision can be done without major modifications of set-up or presentation

**We thank Martin de Kauwe for his constructive comments. A point-by-point response follows below.**

Tschumi et al. explore the timely topic of the impact of combined drought/heat extremes on terrestrial carbon dynamics, using the LPX model. The authors construct a suite of potential forcings and use a factorial approach to examine the follow-through impact on carbon/tree dynamics. I found the paper well written, the methods are well constructed and the analysis broadly interesting. I did feel that more could have been done to explain key assumptions (see below) and in particular to leverage these assumptions in the discussion text when discussing impacts/broadening how transferable the results are across models. I think the equilibrium analysis needs more motivation in the paper, currently, 2 of the 9 figures relate to this and there is no mention in the abstract.

**Thank you for your helpful comments and overall positive feedback.**

The first major issue relates to what LPX assumes, particularly with respect to temperature. The manuscript lacks information on the assumed bioclimatic limits and that photosynthesis and respiration estimates are simply down-regulated if temperature bounds are exceeded (I'm going off what I know about LPJ-GUESS, so please correct me). I think this information should be in the methods (a table by PFT) as it is likely to impact on how transferable the results are across models. For example, how much of the impact of the hot (or hotdry) is due to the assumed upper bounds of the model? Are these sensible in all geographic locations (see Australia, for example, where I also note the model has a continent of grasses - https://bg.copernicus.org/articles/18/2181/2021/). I would anticipate some return to this in the discussion when the authors try to contextualise their results. I want to be clear though, I'm not suggesting this invalidates anything the authors have done, but I would nevertheless anticipate a degree (a lot) of sensitivity to the assumptions of the DGVM. Similarly, some text on how well the model simulates the impact of drought.

**Thank you for this comment. We agree that the results are probably highly model dependent and a model intercomparison is under way. To be a bit more explicit about LPX's assumptions, we have included a table with LPX's specific bioclimatic limits in the appendix. In addition, we have extended the Methods section on model processes and discussed some of the models limitations in the Discussion section.**

**The following text was added to the Methods: "Here, only natural vegetation is considered, which is internally represented by eight tree PFTs and two herbaceous PFTs competing for resources and adhering to bioclimatic limits, which are listed in Table A1 as well as other process parameterizations (e.g. temperature dependence of photosynthesis or water balance). These bioclimatic limits and other parameters as well as process representation can differ from model to model, leading to a different response of the vegetation to extreme climatic events." … "As an example, the bioclimatic parameter governing the upper limit of temperature is implemented in LPX-Bern by inducing mortality proportional to the number of days in the year where this threshold is exceeded.**

**Other models may not only use different values for the threshold and a different relationship between mortality and exceedance, but an altogether different parameterization. This will in turn influence the response to the heat stress in the model."**

**In the Discussion, we added: "Every model has biases and limitations which could be discussed at length, but for argument's sake we will only discuss some of them briefly. One important component in LPX are the bioclimatic limits, as already mentioned in Sect. 2.2. Mortality induced by maximum temperature only affects tropical trees. One could imagine a different extreme response if this parameter also applied for grasses. As it is, C4 grasses are very water-efficient in LPX, which leads to Australia being a bit too green in our simulations compared to observations, as an example. This could also explain why grasses thrive in the *Hotdry* scenario. The parameterization of the water balance is another possible factor that greatly influences the response to dry conditions. LPX has a relatively simple supply and demand driven water limitation and for instance does not consider effects of Xylem damage (Arend et al., 2021). Overall, models may differ strongly depending on model parameterizations and process representations (Paschalis et al., 2020)."**

This point about assumptions can be extended when the authors discuss how competition changes fractional cover with scenarios. I think the C4 grass component of the model will have assumed bioclimatic limits, in some places (Australia I know for certain), these are likely to bias interpretations …

**The Discussion has been extended with some limitations and biases, which include a wet bias in Australia in the EC-Earth forcing data and the fact that our C4 grasses are very water-efficient in LPX, which both leads to Australia being a bit too green in our simulations compared to observations, see also the response above.**

The second major issue relates to missing insight into how realistic the output from EC-Earth is? Is it biased warm/cold, wet/dry. The model will clearly have biases and I'm not suggesting that these biases would warrant the paper not being accepted; however, they do affect the subsequent analysis through LPX. It is really important that some text is added about this I feel. Having personally been recently looking at the mean and variability bias in future precipitation data over Australia, I think this is a significant issue.

**Note that we discussed the biases in EC-Earth in detail in an earlier manuscript: https://rmets.onlinelibrary.wiley.com/doi/full/10.1002/gdj3.129. For completeness, we now extended the section on the EC-Earth bias in the Data section and summarize the main biases. It is important to keep in mind that the analysis was done by looking at changes relative to the control (which has the same mean bias as the rest of the scenarios) and not to observations, therefore the EC-Earth bias does not play a relevant role when analysing the differences between scenarios.**

Finally, I think it would be great to add some figure (timeseries) that shows a case study of one of these very extreme summers to see how the model is behaving, for some region. I might be alone here, but I'd appreciate seeing how the model dynamics in terms of uptake/respiration play out across an extreme scenario vs one without events. It is quite hard to get this message from the maps.

**Thank you for this suggestion. We have tried to come up with a time series illustration that would be useful for the overall story but did not succeed. Generally we look at how changes in the climatology of the occurrence of droughts and heatwaves affect vegetation distribution and carbon dynamics. Hence focusing on single events would not be very informative but rather illustrate the generic response of LPX to certain types of climate extremes. To address the reviewer's comment we have expanded the Introduction to make sure the scope of this study is clear and added some discussion on how LPX deals with droughts and heatwaves. The last paragraph of the introduction now starts with:**

**"In this study, we aim to disentangle the differential effects of different frequencies of hot conditions, dry conditions, and compound hot-dry events on vegetation composition, carbon pools, and carbon dynamics. Our main motivation is to test the sensitivity of a commonly used vegetation model to differences in the climatology of the occurrence of hot and dry extremes and how these changes in drought and heat occurrence affect vegetation distribution and carbon dynamics."**

Minor

- Line 23: Allen et al. isn't an appropriate citation for CO2-induced changes in WUE.

**We have changed this reference to De Kauwe et al. (2013) (Global Change Biology, 19, 1759-1779, https://doi.org/10.1111/gcb.12164).**

- I think the scenarios sound well thought through and I note what I suspect is a deliberate naming choice to call one of the "dry" rather than drought. I think this is the right choice as the authors can't be sure it indeed does sample a drought (unless they check). However, the text surrounding Table 1 talks in terms of "drought". I think it would be better to be consistent with the language of the scenarios (?). I note further on they talk about checking for drought using SPI. Could they perhaps add a few sentences to explain a bit more about how they did this, it is currently unclear for me.

**We have made sure that we always talk about 'dry' conditions instead of 'drought'. For a more in-depth analysis of the characterization of dry events we cross-referenced our previous paper where this is explained in more detail.**

- I thought the equilibrium text was interesting but I'm not sure that the context was really given? If I missed it, I apologise but I would have liked to have seen more in the

intro/methods to explain motivation. I also wonder whether each change in Fig 7 would be better on the same y-axis so the reader can compare relative impacts.

**The equilibrium analysis provides some insight into how quickly the model responds to a sudden change in forcing, here a change in frequency of hot and dry events, and how quickly the model converges to a new equilibrium. We added a sentence to the Methods to motivate this aspect better ("By running the model with two different spin-ups per scenario, we explore the model equilibrium and how fast the model reacts after a step change in the frequency of extreme events."). Regarding Figure 7, we prefer to keep the different y-axes because the results are easier to read this way. See the following version of Figure 7 which has the same y-axes, in this version differences between the scenarios are difficult to assess.**

[Figure]

- With the maps, consider chopping off Antarctica to show more of the land surface.

**We decided to leave Antarctica on the maps and stick to the Robinson projection. Python only allows us to plot the whole world when using the Robinson projection.**

Martin De Kauwe

**We thank Christopher Reyer for his constructive comments. A point-by-point response follows below.**

The paper is very well written, treats an important topic and is certainly an interesting study. I have some main and minor comments which I think can all be accommodated with wither few additional simulations and/or discussion of the results/model.

**Thank you for your positive and helpful comments**.

1) the main results seem to be a 4% change in ecosystem productivity. I think the paper needs a bit more work to carve out why these 4% actually matter, also in light of the many uncertainties. as written now, it seems the number is "just a number" and as such not very impressive since it is "only" 4%. So I think you should carefully check where you can give more depth to this result by either referring to regional patterns, eg. "the 4% may not seem much at global level but in region xxx, forests are strongly reducing productivity" or sth like that and by better placing the results into context, i.e. what do 4% really mean in terms of changing ecosystems/landscapes and other ecosystem functions and processes and in what direction are the uncertainties of the model etc. pushing this finding?

**We agree that these global numbers provide little information on regional patterns. We have therefore extended the abstract and added regional information on the 10% vegetation cover change (based on Fig. 3), 4% ecosystem productivity change (based on Fig. 5) and also extended on this a bit in the Results section.**

**The following sentence was added to the Abstract: "Regionally, this value can be much larger, for example up to -80 % in mid-western U.S. or up to -50 % in mid-Eurasia for *Hotdry* tree ecosystem productivity."**

**In the Results, we also extended on the regional patterns in several paragraphs: "Regionally, this increase can be much larger. Total tree cover for the mid-west of the U.S., for example, is increased by up to 400 % and there is a similarly large increase in South Africa (results not shown). These are regions with nearly no trees in the *Control* scenario (Fig. 1). A smaller, but still large increase of up to 100 % is observed in South America, southern Africa and large parts of Eurasia."**

**"At the regional scale, the decrease is largest in the mid-west of the U.S. with up to -80 % as well as up to -50 % in mid-Eurasia."**

**"..., mainly in the U.S., Europe, mid-Eurasia and southern South America,…"**

**"Relative carbon flux reductions can be much larger for some regions, for example up to -80 % in the mid-west of the U.S., mirroring the decrease in tree cover."**

2) One other main point that deserves more attention: is LPX (or any of such vegetation models) actually up to the challenge of simulating the effects of compound events? I am sure the model has been well-tested for the control and noextremes scenarios but does

it reflect the processes that really matter in a hotdry scenario? e.g. in many regions insect damage following a drought is far more substantial than the drought damage (direct through losses of leaves and branches etc... and indirect through changes in carbon cycling etc.) and hence if the model only incorporates the "physiological" responses (which it is probably still likely do underestimate because of a lack of processes such as xylem embolism in the model etc...) it might underestimate the effects of compound events. I think the manuscript should spell out more clearly which the crucial processes in the model are, if the model has been tested with regard to those and how realistic they are to actually cover the effects of compound events in their full breadth (or at least a large part).

**There are definitely many important aspects that LPX does not include. We have expanded on model limitations that are potentially relevant for hot and dry extremes in the Methods and Discussion sections. Generally a main motivation of the work is to test the sensitivity of an off-the-shelf vegetation model to differences in the climatology of the occurrence of hot and dry extremes, rather than providing realistic simulations of the responses to extreme events. We have clarified this aspect in the Introduction.**

**The Methods have been extended with: "As an example, the bioclimatic parameter governing the upper limit of temperature is implemented in LPX-Bern by inducing mortality proportional to the number of days in the year where this threshold is exceeded. Other models may not only use different values for the threshold and a different relationship between mortality and exceedance, but an altogether different parameterization. This will in turn influence the response to the heat stress in the model."**

**The last paragraph of the Discussion now says: "All results, such as the exact changes in vegetation distribution and carbon uptake, are somewhat sensitive to the choice of the dynamic global vegetation model and the employed climate model. Every model has biases and limitations which could be discussed at length, but for argument's sake we will only discuss some of them briefly. One important component in LPX are the bioclimatic limits, as already mentioned in Sect. 2.2. Mortality induced by maximum temperature only affects tropical trees. One could imagine a different extreme response if this parameter also applied for grasses. As it is, C4 grasses are very water-efficient in LPX, which leads to Australia being a bit too green in our simulations compared to observations, as an example. The parameterization of the water balance is another possible factor that greatly influences the response to dry conditions. LPX has a relatively simple supply and demand driven water limitation and for instance does not consider effects of Xylem damage. Overall, models may differ strongly depending on model parameterizations and process representations (Paschalis et al., 2020). Furthermore, some uncertainties also arise from the model setup. For example, land-atmosphere feedbacks may play an important role (Humphrey et al., 2021), which are not considered in such an offline model setup as we have conducted in this study. Considering the number of uncertainties that may govern vegetation**

**and carbon cycle response to varying drought-heat signatures, a model intercomparison project using our scenarios as forcings for different vegetation models has already been set up and may reveal insights on how model differences affect the results."**

**The Introduction now includes the following sentence: "Our main motivation is to test the sensitivity of a commonly used vegetation model to differences in the climatology of the occurrence of hot and dry extremes and how these changes in drought and heat occurrence affect vegetation distribution and carbon dynamics."**

3) the role of CO2 needs clarification. in your experimental set-up you keep co2-contant which is fine. but through the stomatal responses water-use efficiency might be higher under higher co2 under which such compound events might occur. I know this introduces another dimension into your experimental set-up which you may not be able to accommodate but I think the issue is at least worthwhile further discussion.

**Yes, CO2 fertilization is not considered in this study. We have now mentioned in the Discussion that this is an effect that potentially has a large impact in a changing climate. Again, note that this study is primarily a sensitivity analysis of modeled vegetation to varying frequencies in droughts and heatwaves, keeping everything else constant.**

**We added the following paragraph to the Discussion: "We run the vegetation model offline, that is, there is no feedback from the land surface to the climate, and keeping the atmospheric CO2 level constant. Processes in the real world might be more complex. Especially CO2 fertilization, where higher CO2 concentrations lead to a more efficient uptake of CO2 by the plants and thus less chance to lose water through open stomata, may modulate how hot and dry conditions affect vegetation and carbon dynamics in the future (Domec et al., 2017, DeKauwe et al., 2021)."**

minor comments:

L34: what is the observational period here?

**The observational period is 1982-2016. We have added this information to line 34.**

L198ff: you should discuss later whether you expect this equilibrium to be reached soon or whether there is a need to lengthen the simulation time.

**This analysis illustrates the response time of LPX to a step change in extreme event occurrence (referred to as "shared spin-up"). For the main analysis, however, we used the simulations that used each scenario forcing already in the spin-up and were in equilibrium for the analysis period (referred to as "individual spin-up"). We have added some additional information to clarify this point.**

Christopher Reyer

---

## Author Response (AR2)

**We thank Christopher Reyer for his comments. We have included his suggestions to improve our manuscript and added a point-by-point response here.**

L33ff. I understand your focus on drought and heat here and I do not want to downplay their importance but one key issue that I think is often overlooked is that these extremes interact and predispose other, more ecological extremes such as forest fire and insect outbreaks. in this paper (10.1038/NCLIMATE3303) on figure 2b we show that drought has in most cases found in the literature an amplifying effect on other forest disturbances. No need to cite this particular paper, I am sure you know similar ones, but I think the point for follow-up extremes and interacting disturbances could be made here (and extend the compound event concept).

**We agree and now acknowledge that other extremes such as fire or insect outbreaks can interact with droughts and heatwaves:**
**"In many cases, drought and heat predispose or interact with other hazards and disturbances such as forest fires and insect outbreaks (Seidl et al., 2017)."**

L170ff: you could also mention here what happens to NPP if trees replace grasses or vice versa

**We have added a few sentences to this topic:**
**"The response of NPP to the replacement of trees with grasses and vice versa is varied, as it strongly depends on environmental conditions and vegetation composition. Generally, NPP is greater for trees than for grasses, which implies that global NPP is larger in a world with more trees and smaller if more forest area is replaced by grassland."**
**"Interestingly, although grass cover is increased in the *Hot* scenario (Fig. 2a), NPP in grasslands is reduced (Fig. 2b), explaining the lack of change in global NPP for the *Hot* scenario (Fig. 4a)."**

L246ff if the models separates between c3 and c4 grasses you could discuss here the differential effects of co2 and how this would influence your grassland results if you had included co2. NOTE: I now saw that you mention c4 grasses in L301 so maybe a discussion is better place there. your call.

**We have added a sentence on the effect of CO2 on C4 grasses in the discussion:**
**"A potential increase in atmospheric CO2 conditions as it is predicted by socio-economic scenarios would further alleviate drought stress and thus benefit C4 grasses."**

L266ff: you could contrast your results on low effects on tropical trees here with the strong effects recent droughts had on the amazon forest

**We now mention other studies that have found an effect of droughts on the amazon forest:**

**"However, case studies on recent droughts in the Amazon forest show how tropical forests can be negatively affected by drought conditions (Doughty et al., 2015, Feldpausch et al., 2016, Machado-Silva et al., 2021)."**

Table A1: I think it would be good to give the full names of the PFTs, I could not really find them anywhere else and although I can guess what they mean for most of them, it would be easier to have them in table A1.

**We have included the full names of the PFTs in Table A1.**